

# Differential expression of lysine acetylation proteins in gastric cancer treated with a new antitumor agent bioactive peptide chelate selenium

Yanan Xu[1,*], Jianxun Wen[2,*], Wenyan Han[3], Jin Yan[1], Wei Jia[1] and Xiulan Su[1]

[1] Clinical Medical Research Center, Inner Mongolia Bioactive Peptide Engineering Laboratory, The Affiliated Hospital, Inner Mongolia Medical University, Hohhot, China
[2] College of Basic Medicine, Inner Mongolia Medical University, Hohhot, China
[3] Clinical Laboratory, The Second Affiliated Hospital, Inner Mongolia Medical University, Hohhot, China
[*] These authors contributed equally to this work.

## ABSTRACT

The method of anticancer bioactive peptide (ACBP) functionalized selenium particle (Se), which has enhanced anticancer activity, inhibited the growth of gastric cancer (GC) cells, and increased the ability of apoptosis *in vitro*, has been reported in previous studies. We used tandem mass spectrometry (TMT) labeling to construct a complete atlas of the acetylation-modified proteome in GC MKN-45 cells treated with ACBP-Se. The proteomics data database was searched and analyzed by bioinformatics: Kyoto Encyclopedia of Genes and Genomes (KEGG), Gene Ontology (GO), functional enrichment, and protein-protein interaction network. Finally, we conducted a quantitative PRM analysis of the selected target-modified peptides. We identified 4,958 acetylation sites from 1,926 proteins in this research. Among these, 4,467 acetylation sites corresponding to 1,777 proteins were quantified. Based on the above data and standards, we found that in the ACBP-Se group vs. the control group, 297 sites were upregulated, and 665 sites were downregulated. We systematically assessed the proteins containing quantitative information sites, including protein annotation, functional classification, and functional enrichment, cluster analysis supported by functional enrichment, domain structures, and protein interaction networks. Finally, we evaluated differentially expressed lysine acetylation sites. We revealed that SHMT2 K200 and PGK1 K97 were the most critical acetylated non-histone proteins, which may have an essential role in ACBP-Se treatment. Here, we identified and quantified the lysine acetylation proteins in GC cells treated with ACBP-Se. The characterization of acetylation indicates that acetylated proteins might be pivotal in the biological process, molecular binding, and metabolic pathways of ACBP-Se treatment progress. Our findings provide a broad understanding of acetylation ACBP-Se treatment of GC, suggesting a potential application for molecular targeted therapy.

Corresponding author
Xiulan Su, xlsu2014@163.com

## INTRODUCTION

Gastric cancer (GC) is a malignant digestive tract tumor that seriously affects human health (*Russo & Strong, 2018*). The GC etiology is complex, and the clinical symptoms are atypical. In the past 30 years, with improvements in people's living conditions and early GC prevention, the incidence and mortality of GC have been limited worldwide. GC still poses a severe threat to human health. The study demonstrated that the incidence of GC is close to that of lung, breast, rectal, and prostate cancer. Among all cancers, GC is the second leading cause of death and significantly burdens the medical system (*Strong, 2020*).

Despite continuous advances in biotechnology and medical services in recent years, the incidence of GC is still increasing, and the affected population is gradually becoming younger. Therefore, finding an effective treatment for GC is extremely urgent and necessary.

Selenium (Se) is an essential trace element in organisms. Se can improve the normal function of the redox system, immune system, and anticancer effects (*Arthur, McKenzie & Beckett, 2003*). Se can also enhance the ability of the immune system to recognize viruses, foreign bodies, diseases *in vivo*, promote the proliferation of immune system cells, the synthesis of antibodies, increase the level of antibodies in the blood, and enhance the phagocytic function of the immune system cells (*Achilli, Ciana & Minetti, 2018*; *Avery & Hoffmann, 2018*). The nutritional status of Se is closely related to cancer, which can remarkably control the deterioration of the disease, reduce the side effects of chemotherapy and radiotherapy, and promote the efficacy of anticancer drugs (*Vinceti et al., 2018*). A more exciting research area in drug design is the synthesis of peptide-Se chelates, which have additional advantages in inhibiting cancer stem cell lines (*Li et al., 2021*). Anticancer bioactive peptide (ACBP) is a low molecular weight active substance extracted from goat livers. ACBP shows excellent antitumor activity and can inhibit tumor growth in nude mice (*Yu et al., 2014*). ACBP increases the sensitivity of chemotherapeutic drugs and reduces the side effects of chemotherapy (*Su et al., 2013*).

Here, with ACBP sulfhydrylation, many sulfhydryl groups on the molecule serve as binding sites for Se. ACBP-Se has been previously found to have a high susceptibility to GC cells, inhibit their growth, and increase their ability for cell apoptosis in vitro. Therefore, ACBP-Se may become an ideal candidate for treating GC (*Li et al., 2021*).

Evidence has shown that protein acetylation might be a prospective candidate for the targeted pathway in cancer treatment. Epigenetic changes lead to the differential modification of gene expression, which is closely related to the occurrence and development of GC (*Gil, Ramirez-Torres & Encarnacion-Guevara, 2017*). Protein acetylation, displayed in histone and non-histone proteins, such as transcription factors, has been demonstrated to have a significant role in the epigenetic regulation of various cancers and exhibits high tissue specificity (*Christensen et al., 2019*).

In GC research, non-histone mutation or abnormal expression of non-histone deacetylase may become a potential target of targeted therapy for GC treatment. However, there are limited studies on the acetylation targets of ACBP-Se in GC treatment. The principal reason is that the critical point of protein acetylation and the location of the acetylation protein in GC remain unclear.

Proteomic methods are based primarily on mass spectrometry (MS), with high specificity and sensitivity benefits. In this study, we quantitatively analyzed the changes in non-histone protein acetylation modification in GC cells treated with ACBP-Se using tandem mass spectrometry (TMT) labeling and mass spectrometry-based quantitative proteomics. Our objective is to provide a novel target, and therapeutic scheme for ACBP-Se targeted GC therapy.

## MATERIAL AND METHODS

### Cell culture and cell treatment

The human GC cell line MKN-45was purchased from the Cell Resource Center, Institute of Basic Medical Sciences, Chinese Academy of Sciences, Peking Union Medical College. Cell culture was done at the Clinical Medical Research Center of Inner Mongolia Medical University. S-acetylmercaptosuccinic anhydride (S-AMSA) and hydroxylamine hydrochloride were purchased from Sigma. The MKN-45 cells were cultured in RPMI-1640 media (Invitrogen; Thermo Fisher Scientific, Inc., Waltham, MA, USA) with 10% fetal bovine serum (FBS; HyClone; GE Healthcare Life Sciences, Marlborough, MA, USA) and 1% penicillin-streptomycin (Invitrogen; Thermo Fisher Scientific, Inc., Waltham, MA, USA). The cells were maintained in a humidified $CO_2$ incubator at 37 °C. ACBP-Se extraction and purification were conducted as previously reported. The yield of MKN-45 cells cultured in the laboratory was $1 \times 10^6$ cells/mL. After 24 h of culture, 5 mg/mL of induced ACBP-Se was added to the ACBP-Se group (S). Equal doses of PBS were added to the control group (C).

### Protein extraction

The samples were maintained on ice in lysis buffer (8M urea, 1% protease inhibitor cocktail). The remaining debris was removed by centrifugation at 12,000 g for 10 min at 4 °C. The protein concentration was measured using a BCA kit.

### Trypsin digestion

The protein sample was reduced with 5 mM dithiothreitol for 30 min at 56 °C and alkylated with 11 mM iodoacetamide for 15 min. The protein sample added 100 mM TEAB to a urea concentration of less than 2M. Trypsin was added at a mass ratio of 1:50 trypsin to protein for digestion overnight and a mass ratio of 1:100 trypsin to protein for 4 h digestion.

### HPLC fractionation

The trypsin peptides were separated into fractions by high pH reverse-phase high-performance liquid chromatography (HPLC) using the Thermo Betasil C18 column. Finally, the sample was combined into six fractions and dried by vacuum centrifuging.

### LC-MS/MS analysis

The trypsin peptides were resolved in 0.1% formic acid (solvent A) and directly loaded onto a self-made reversed-phase analytical column. The gradient increased from 6% to 23% with 0.1% formic acid in 98% acetonitrile (solvent B) over 26 min. The sample peptides were treated with an NSI source, followed by tandem mass spectrometry (MS/MS) in Q Exactive TM Plus (Thermo) coupled online to the UPLC.

## Database search

The resulting MS/MS data were processed using the Max quant search engine (v.1.5.2.8).

## Bioinformatics methods

### Annotation methods

Gene ontology (GO) proteome annotation was built on the UniProt-GOA database (http://www.ebi.ac.uk/GOA/). Then, proteins are classifiable by GO annotation based on three categories: biological process, cellular component, and molecular function. Identified protein domain functional descriptions are annotated by InterPro Scan (a sequence analysis application) built on the protein sequence alignment method affiliated with the InterPro (http://www.ebi.ac.uk/interpro/) domain database. InterPro is a database that integrates diverse information about protein families. The Kyoto Encyclopedia of Genes and Genomes (KEGG) database was used to annotate the protein pathways.

## Functional enrichment

A corrected $p$-value lower than 0.05 was considered significant in the GO analysis. A corrected $p$-value lower than 0.05 was considered significant in the KEGG database pathway. A corrected $p$-value lower than 0.05 was considered significant in the protein domains.

## Protein-protein interaction network

All differentially expressed modified protein database sequences were searched against the string database version 10.1 for protein-protein interactions.

## PRM protein extraction

Triton-100 1%, protease inhibitor1%, dithiothreitol 10 mM and urea 8M used high-intensity ultrasonic processor sonication three times and 20,000 g at 4 °C for 10 min. The protein adds 20% TCA for 2 h at −20 °C. The protein was rediscovered in 8 M urea. The protein concentration was determined with a BCA kit according to the manufacturer's instructions.

## LC-MS/MS analysis

The sample peptides were resolved in 0.1% formic acid. The gradient comprised an increase from 6% to 23% solvent B (0.1% formic acid in 98% acetonitrile) over 38 min, 23% to 35% in 14 min, and climbing to 80% in 4 min, then holding at 80% for the last 4 min, all at a constant flow rate of 700 nL/min on an EASY-nLC 1000 UPLC system. The sample peptides were subjected to an NSI source and were followed by tandem mass spectrometry (MS/MS) coupled online to the UPLC. Peptides were then selected for MS/MS using an NCE setting of 27. The fragments were identified in the Orbitrap at a resolution of 17,500.

## Statistical analysis

The resulting MS data were processed using Skyline (v.3.6; https://skyline.ms/project/home/software/Skyline/begin.view?). Peptide setting as Trypsin [KR/P], max missed cleavage set as 2. The peptide length was set at 8–25. The variable modification was set as carbamidomethyl of Cys and oxidation on Met, and the maximum variable modifications were set at 3.

## RESULTS

### LC MS/MS

The entire experimental procedure is shown in Fig. 1. The modified quantitative principal component analysis results for all samples are shown in Fig. S1. The degree of aggregation between repeated samples in the figure indicates that the quantitative repeatability meets the standard. Figure S2 is a box plot of the RSD of the modified quantitative values among the repeated samples. The quantitative repeatability of the experiment met the standard, the mass error of all the identified peptides. The mass error distribution is close to zero, and mostareb2 ppm, which means that the mass accuracy of the MS data fits the requirement (Fig. S3). Second, the lengths of most peptides were distributed between seven and 13, which accept the property of trypsin peptides (Fig. S4), and means that the sample preparation reached the standard.

### Quantification overview of acetylation sites and proteins

Differentially expressed lysine acetylation sites were also annotated when the criteria of N1.5 or b0.666 were up or downregulated by the Lysacetylation sites, as indicated in Table S1. In this research, we identified 4,958 acetylation sites from 1,926 proteins. Among these 4,467 acetylation sites corresponding to 1,777 proteins were quantified. Our results showed that 655 acetylation sites of 499 proteins were downregulated in the ACBP-Se group compared to the control group. At the same time, 297 acetylation sites of 238 proteins were unregulated (Fig. 2).

### GO functional enrichment

We classified the acetylated proteins into different groups based on cell components, molecular functions, and biological processes to gain functional insight into how Lys acetylation may regulate cellular function. The cell components of acetylated proteins were analyzed using GO enrichment. The acetylated proteins distributed in various kinds of locations, mainly in the cytoplasm (37.12%), nucleus (30.5%), and mitochondria (13.09%) (Fig. 3), were revealed in the results. Furthermore, in detail, low-expression or over-expression of acetylated proteins in ACBP-Se group compared to control group distributed also primarily in extracellular matrix ($p = 0.0001022$), cell cortex ($p = 0.0057799$), endoplasmic reticulum chaperone complex ($p = 0.0104276$), brush border ($p = 0.0127517$), pore complex ($p = 0.0167173$), cluster of actin-based cell projections ($p = 0.0174801$), sarcomere ($p = 0.0223746$), contractile fiber ($p = 0.023869$) (Fig. 4A, Table S2).

The molecular function analysis showed that there were 499 acetylated proteins, mainly in actin filament binding ($p = 0.0011559$), rRNA binding ($p = 0.0041598$), enhancer binding ($p = 0.0061574$), chemoattractant activity ($p = 0.0062698$), anion channel activity ($p = 0.0062698$), voltage-gated ion channel activity ($p = 0.0062698$), voltage-gated channel activity ($p = 0.0062698$), and carbon–carbon lyase activity ($p = 0.0083035$) (Fig. 4B, Table S2). As far as biological process is concerned, the acetylated proteins are allocated in cell morphogenesis ($p = 0.0005497$), cell morphogenesis involved in differentiation ($p = 0.000623339$), positive chemotaxis ($p = 0.000761514$), response to

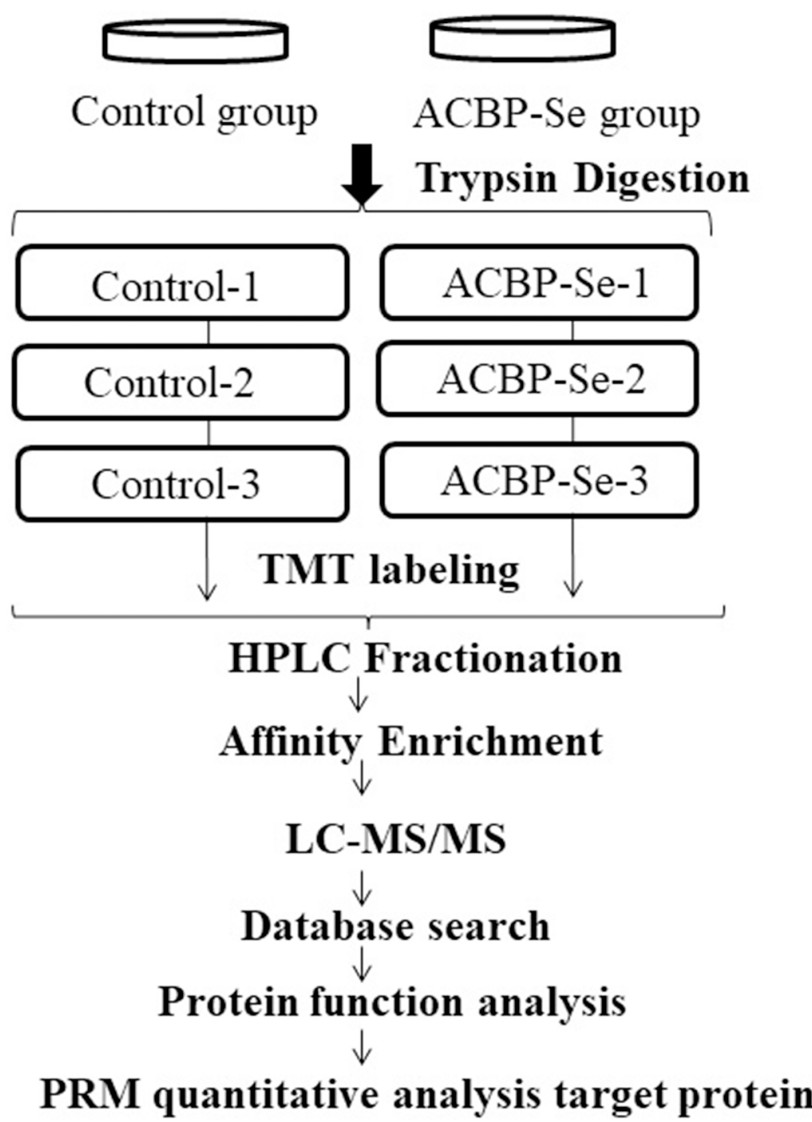

**Figure 1  The entire experimental procedure.** Experimental design for the quantitative proteomic analysis, the experiment was divided into two groups (control, ACBP-Se), and proteins were extracted from three independent biological replicates per treatment. Extracted proteins were prepared labeled with TMT regents. The labeled peptides were separated by HPLC fractionation, and fractions were analyzed by reversed-phase LC-MS/MS.

unfolded protein ($p = 0.001404866$), skeletal system development ($p = 0.001592112$), hair cycle process ($p = 0.001671425$), bone development ($p = 0.001947448$), limb development ($p = 0.002467039$), epigenetic ($p = 0.003168168$), skin development ($p = 0.003893076$), blood coagulation ($p = 0.004312394$), heart development ($p = 0.004394278$), telomere organization ($p = 0.007286699$), and tube morphogenesis ($p = 0.010163738$) (Fig. 4C, Table S2).

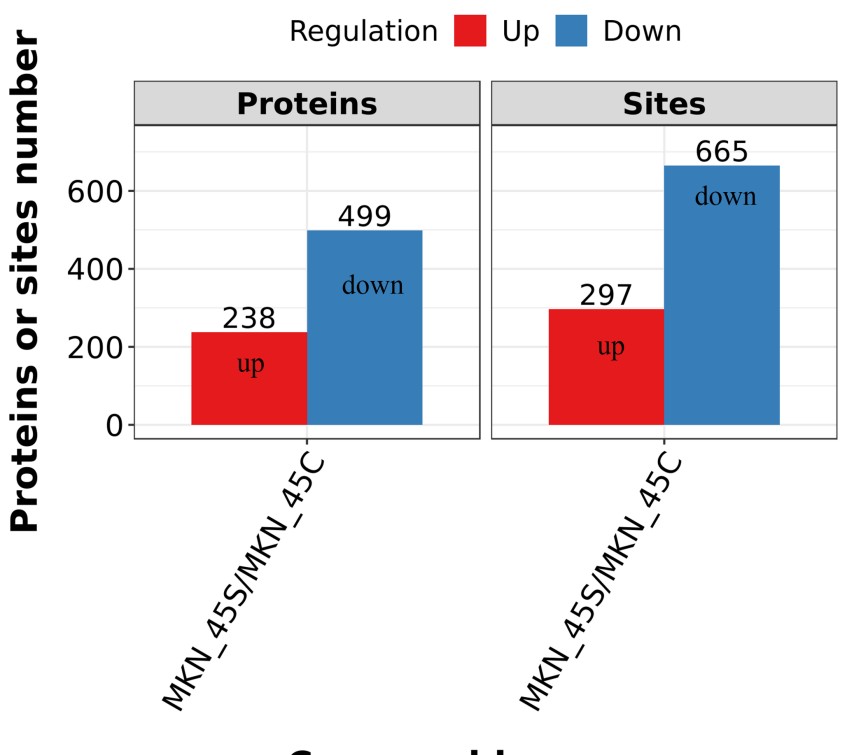

**Figure 2** **Quantification overview of acetylated sites and proteins.** The results identified 4,958 acetylation sites from 1,926 proteins, among which 4,467 acetylation sites corresponding to 1,777 proteins were quantified. The results showed that 655 acetylated sites of 499 proteins were downregulated in ACBP-Se group compared to that in control, while 297 acetylated sites of 238 proteins were unregulated.

## KEGG enrichment

We performed an enrichment analysis of KEGG pathways to understand this cellular pathway involving ACBP-Se compared to the control group. Our data showed that acetylated proteins were widely involved in signaling pathways, including focal adhesion ($p = 0.0050557$), human papillomavirus infection ($p = 0.0054812$), prostate cancer ($p = 0.0080365$), synthesis and degradation of ketone bodies ($p = 0.0174774$), human T-cell leukemia virus 1 infection ($p = 0.028866$), TGF-beta signaling pathway ($p = 0.0306822$), thyroid hormone synthesis ($p = 0.0320377$), arrhythmogenic right ventricular cardiomyopathy (ARVC) ($p = 0.0320377$), butanoate metabolism ($p = 0.0321206$), N-Glycan biosynthesis ($p = 0.0321206$), Th17 cell differentiation ($p = 0.0321206$), mTOR signaling pathway ($p = 0.0379171$), adherens junction ($p = 0.0451676$) (Fig. 5, Table S3).

## Clustering analysis of the Lys acetylation data sets

We divided the Lys acetylation sites with different modification levels into four parts according to their different modification multiples, called Q1 to Q4, as shown in Fig. 6. We then conducted enriched GO, KEGG, and protein domain analyses for each group and cluster analyses to determine the functional correlation of distinct modification multiples.

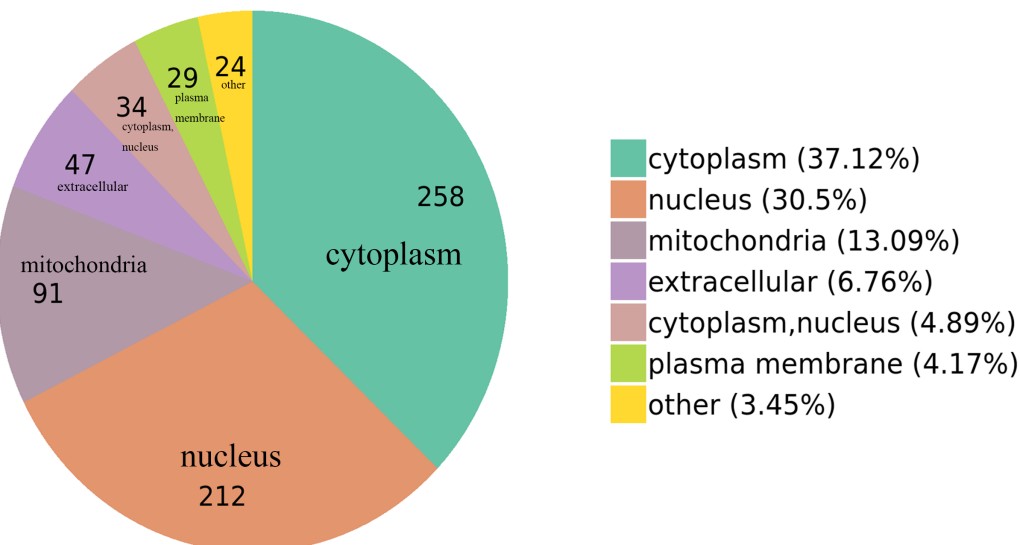

**Figure 3** **GO functional enrichment of locations.** The cell component of acetylated proteins was analyzed by GO enrichment distributed in various kinds of locations, mainly in cytoplasm (37.12%), nucleus (30.5%), and mitochondria (13.09%).

We focused on the acetylation that reproducibly changes within ACBP-Se treatments. We tested the acetylation site data for enrichment in three GO categories, biological process, cell component, and molecular function, to further elucidate the cellular role in ACBP-Se treatments. In the biological process category, processes associated with regulating the response to protein activation cascade ($p = 0.00001$) and cytoplasmic sequestering of protein ($p = 0.0002$) are significantly enriched in the upregulated protein cluster. In contrast, the meiotic cell cycle process ($p = 0.0001687$) and cell morphogenesis involved in differentiation ($p = 0.00209$) were enriched in downregulated proteins (Fig. 7A). It was revealed in the analysis of cellular components that acetylated proteins are enriched mainly in the extracellular matrix ($p = 0.000006$) and cell surface ($p = 0.0002932$) in the upregulated proteins, while the extracellular matrix ($p = 0.0027847$), and nucleoid ($p = 0.0002257$) in the downregulated ones (Fig. 7B). Furthermore, it was shown in the evaluation of a molecular function that proteins were involved in single-stranded DNA binding ($p = 0.0002191$) and nucleic acid binding transcription factor activity ($p = 0.001937$). At the same time, actin filament binding ($p = 0.000007$) and histone threonine kinase activity ($p = 0.0034164$) were in the downregulated protein cluster (Fig. 7C, Table S4).

Protein functions are largely dependent on specific domain structures in the sequence. We performed a domain enrichment analysis to assess the domain structures most regulated by ACBP-Se. We revealed that protein domains were involved in the Concanaval in A-like lectin/glucanase domain ($p = 0.00648309$), Thiolase-like ($p = 0.013724479$), Band 7 domain ($p = 0.020388326$), translation protein, beta-barrel domain ($p = 0.022018744$), B30.2/SPRY domain ($p = 0.037364623$), SPRY domain ($p = 0.037364623$), heat shock protein 70kD, peptide-binding domain ($p = 0.037364623$), heat shock protein 70Kd,

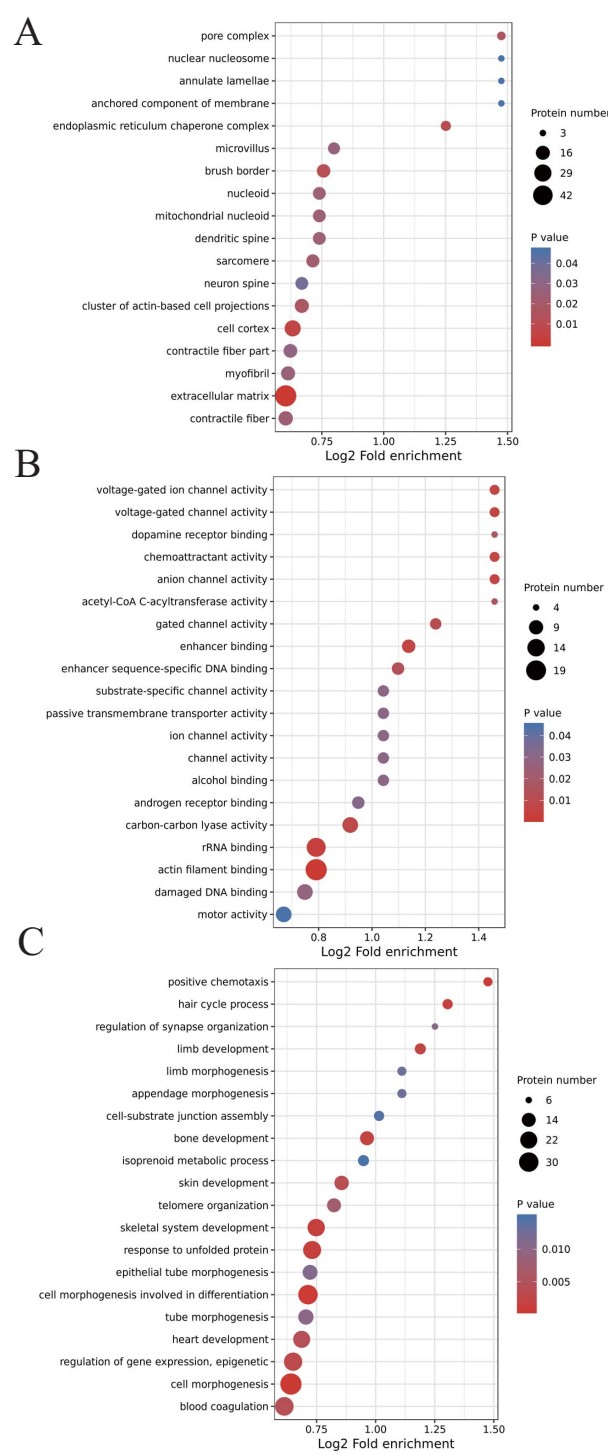

**Figure 4** **GO analyses of low-expression or over-expression of acetylated proteins in ACBP-Se group compared to control group.** (A) The cell component of acetylated proteins also primarily in extracellular matrix, cell cortex, endoplasmic reticulum chaperone complex, brush border, pore complex, cluster of actin-based cell projections, sarcomere, contractile fiber. (continued on next page…)

**Figure 4 (…continued)**
(B) The molecular function analysis showed that of 499 acetylated proteins mainly in actin filament binding, rRNA binding, enhancer binding, chemoattractant activity, anion channel activity, voltage-gated ion channel activity, voltage-gated channel activity, carbon–carbon lyase activity, *et al.* (C) The biological process are allocated in cell morphogenesis, cell morphogenesis involved in differentiation, positive chemotaxis, response to unfolded protein, skeletal system development, hair cycle process, bone development, limb development, regulation of gene expression, epigenetic, skin development, blood coagulation, heart development, telomere organization, tube morphogenesis, *et al.*

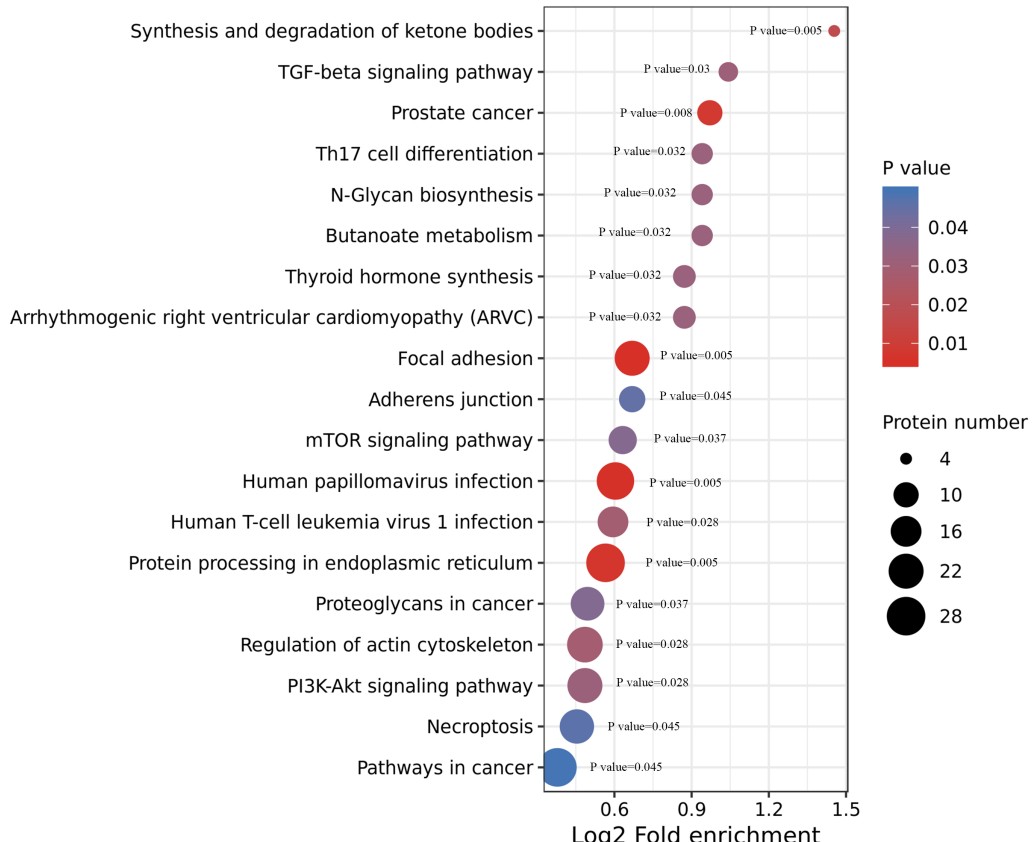

**Figure 5   KEGG enrichment of ACBP-Se group compare to the control group.** KEGG pathways enrichment analysis of ACBP-Se compare to the control group showed that acetylated proteins were widely involved in signaling pathways, including focal adhesion, human papillomavirus infection, prostate cancer, synthesis and degradation of ketone bodies, human T-cell leukemia virus 1 infection, TGF-beta signaling pathway, thyroid hormone synthesis, arrhythmogenic right ventricular cardiomyopathy (ARVC), butanoate metabolism, N-Glycan biosynthesis, Th17 cell differentiation, mTOR signaling pathway, adherens junction.

C-terminal domain ($p = 0.037364623$), and a high mobility group box domain ($p = 0.040640292$) (Fig. 8A, Table S5).

We performed a pathway clustering analysis using KEGG to identify the cellular pathways playing a critical role in ACBP-Se treatments. It was shown by our data that protein processing in the endoplasmic reticulum ($p = 0.0004605$) and complement and

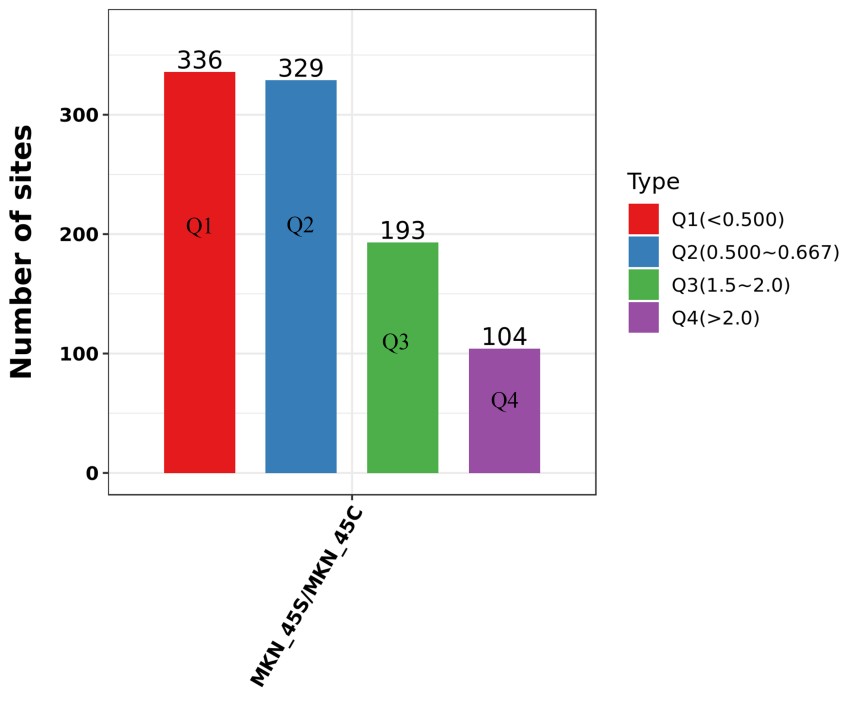

**Comparable groups**

**Figure 6 Clustering analysis of the Lys acetylation.** Lys acetylation sites with different modification divide into four parts according to their different modification multiples, as Q1 to Q4.

coagulation cascade enrichment ($p = 0.00000008$) in upregulation proteins. At the same time, leucine and isoleucine degradation ($p = 0.0002729$) and Th17 cell differentiation ($p = 0.0013603$) were in the downregulated protein cluster (Fig. 8B; Table S6).

## Protein interaction networks of Lys acetylation proteome

Based on the string database, we visualized the protein-protein interaction networks of 243 Lys acetylation proteins. An insight into the probability of the interactions of acetylated proteins in ACBP-Se treatment was offered by our data. A represent example is shown in Fig. 9. Using the MCODE tool, we identified some highly-connected subnet works among ATP-dependent RNA helicase DDX5 (DDX5), serine hydroxymethyl-transferase (SHMT2), phosphoglycerate kinase 1 (PGK1), and protein disulfideisomerase A3 (PDIA3). A high degree of connectivity may be more indicative of a protein complex. Through comparison between the ACBP-Se and control groups, we demonstrated thatpoly (rC)-binding protein 1 (PCBP1), SHMT2, Integrin beta-4 (ITGB4), PGK1, and PDIA3 exhibited a high degree of connectivity located at the center of the network.

## PRM analysis of Lys acetylation proteome

In this research, we performed a quantitative analysis of the PRM for the selected target-modified peptides. We quantified 15 modified peptides, which were consistent with the acetylation group consequences. The results shown in Table 1 of the PRM were quantified by
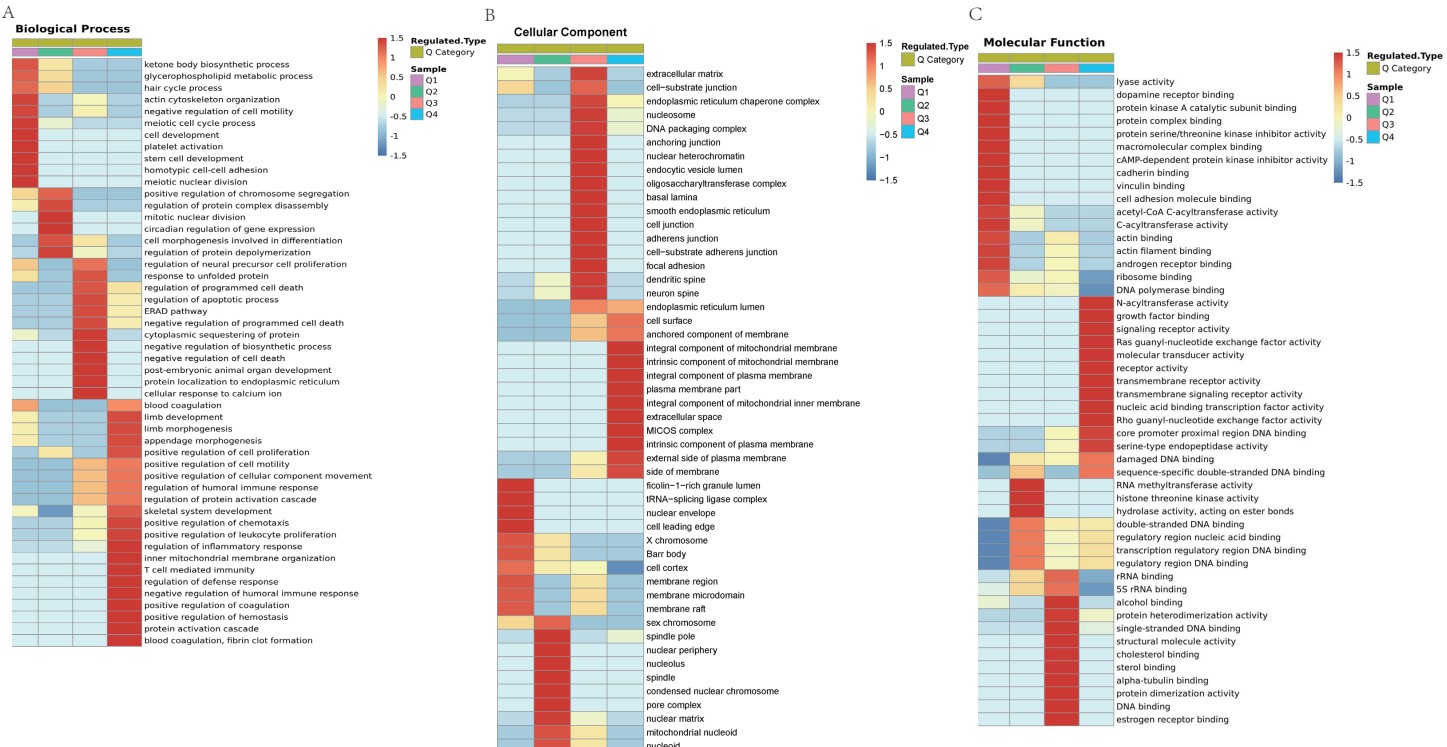

**Figure 7** **GO analyses Lys acetylation data sets enrichment in ACBP-Se group compare to the control group.** (A) Biological process of Lys acety-lation data sets enrichment in ACBP-Se group compare to the control group, processes associated with regulation of response to protein activation cascade and cytoplasmic sequestering of protein are significantly enriched in up-regulated protein cluster. In contrast, meiotic cell cycle process and cell morphogenesis involved in differentiation are enriched in downregulated proteins. (B) The cellular component enrichment results showed that Lys acetylation proteins sets are enriched mainly in the extracellular matrix and cell surface in the up-regulated proteins while extracellular matrix and nucleoid in the downregulated proteins. (C) The molecular function enrichment results showed that Lys acetylation proteins sets are mainly in single-stranded DNA binding and nucleic acid binding transcription factor activity, while actin filament binding and histone threonine kinase activ-ity in down-regulated protein cluster.

peak area. The results showed that the SHMT2 (p34897) acetylation modified differential peptide (LNPK(Ac)TGLIDYNQLALTAR) and PGK1 (p00558) acetylation modified differential peptide (DVLFLK(Ac)DCVGPEVEK) were consistent with acetylation modified proteomics.

## DISCUSSION

The acetylation of lysine was first discovered in 1964. In recent decades, the post-translational modification of lysine acetylation has played an essential role in regulating histone and non-histone functions. Post translational modification (PTM) is an important epigenetic mechanism that regulates biological processes. Research has found that phosphorylation, acetylation, ubiquitination, succinylation, crotonylation and methylation are closely related to the occurrence and development of gastric cancer (*Gil, Ramirez-Torres & Encarnacion-Guevara, 2017*). Non-histone acetylation modification affects protein function by regulating protein stability, enzyme activity, subcellular localization, cell

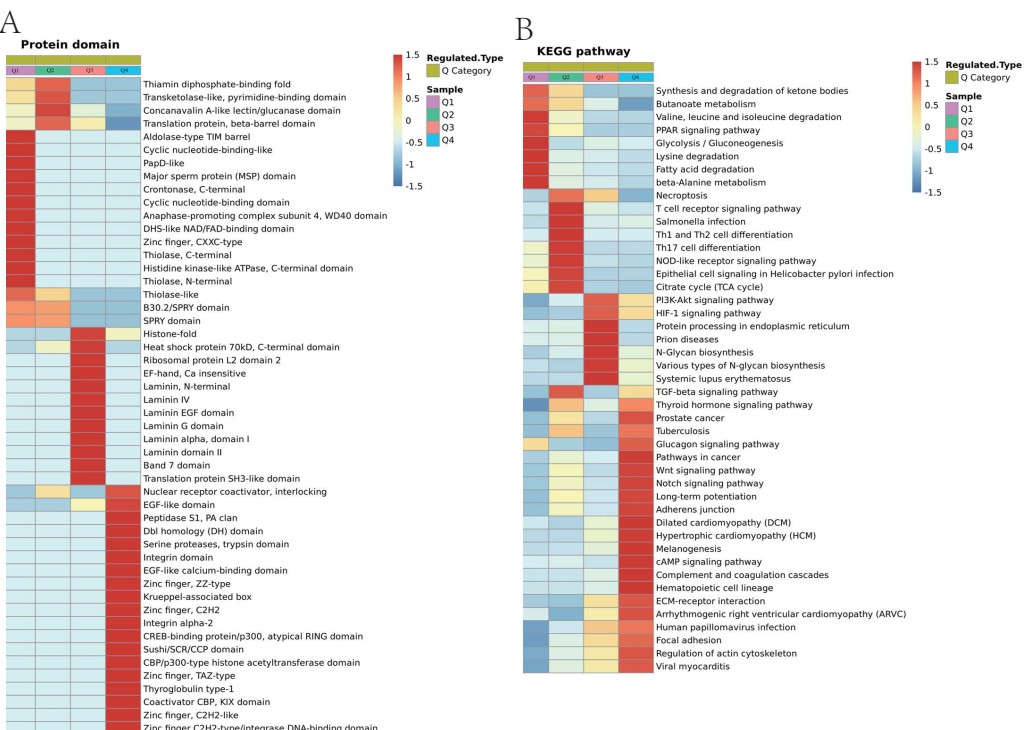

**Figure 8** **Domain enrichment analysis of Lys acetylation data sets in ACBP-Se group compare to the control group and KEGG pathway clustering identify cellular pathways playing role in ACBP-Se treatments.** (A) The results of domain enrichment analysis showed that protein domains involved in Concanavalin A-like lectin/glucanase domain, Thiolase-like, Band 7 domain, translation protein, beta-barrel domain, B30.2/SPRY domain, SPRY domain, heat shock protein 70kD, peptide-binding domain, heat shock protein 70kD, C-terminal domain, High mobility group box domain. (B) KEGG pathway clustering identify cellular pathways playing role in ACBP-Se treatments. The KEGG pathway clustering analysis results showed that protein processing in endoplasmic reticulum and complement and coagulation cascades enrichment in up-regulation proteins, while leucine and isoleucine degradation and Th17 cell differentiation in down-regulated protein cluster.

division, signal transduction, protein folding, autophagy, etc. Non-histone acetylation is related to gene activation, and involved in the proliferation, apoptosis, migration and invasion and regulation of genetic information of gastric cancer cells (*Christensen et al., 2019*). However, due to the limitations of protein acetylation technology, the number of differential acetylation modification sites in GC has exceeded expectations. Therefore, there is an urgent need to study the specificity and specific modification of acetylated proteins for targeting therapy (*Filippakopoulos & Knapp, 2014*).

GC is a significant threat to human health worldwide (*Smyth et al., 2020*). Se is part of the essential trace element that can fundamentally improve the ability to resist diseases in the human body. Se compounds can be used as adjuvant therapies for cancer. It can become an internal environment that inhibits the division and proliferation of cancer cells (*De Oliveira et al., 2020*; *Carlisle et al., 2020*). It has been shown in previous studies that the combination of anticancer bioactive peptides (ACBP) and oxaliplatin (OXA) can significantly depress the growth of gastric cancer-cellMKN-45, promote the apoptosis of

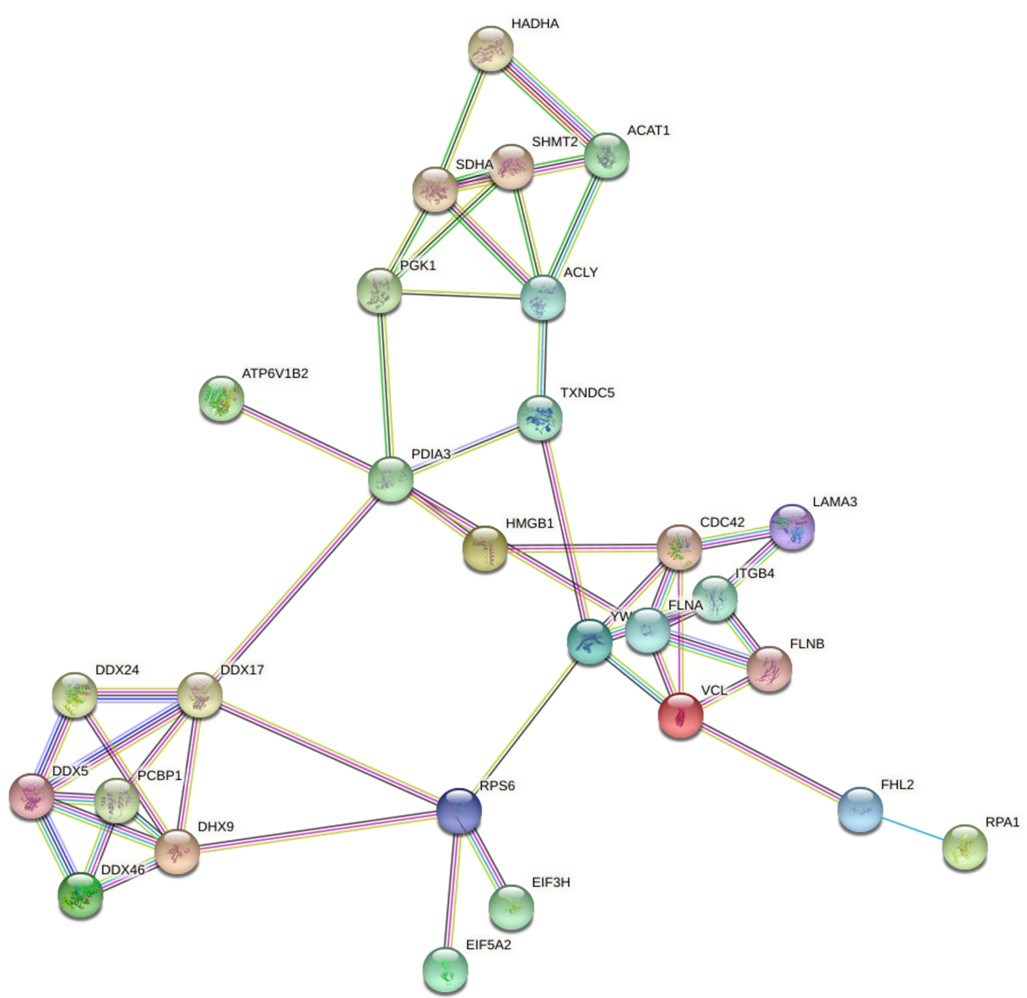

**Figure 9** **Protein interaction networks of Lys acetylation proteome.** Using the MCODE tool, through an intergroup comparison between the ACBP-Se and control groups demonstrated that PCBP1 (Poly (rC)-binding protein 1), SHMT2, Integrin beta-4 (ITGB4), PGK1, PDIA3 exhibited a high degree of connectivity, which located at the center of the network.

MKN-45, and lead to an irreversible arrest in the G2/M phase of the MKN-45 cell cycle (*Li et al., 2017*). Here, the polypeptides were modified by sulfhydrylation to combine Se to form the polypeptide-Se chelate, where the MKN-45 cell line acetylation was investigated.

We used quantitative acetylation proteomics to achieve a global view of acetylation in ACBP-Se treatment of GC. Our results showed that 655 acetylation sites of 499 proteins were downregulated in the ACBP-Se group compared to the control group, while 297 acetylation sites of 238 proteins were unregulated. This result reveals the profile of acetylated proteins involved in ACBP-Se treatment.

We utilized GO annotation to analyze the probable functions of the acetylated proteins and further investigate the characterization of these identified acetylated proteins. It was shown in our results that the acetylated proteins of ectopic expression in ACBP-Se

**Table 1  PRM analysis of Lys acetylation proteome.**

| Protein Accession | Modified Sequence | MKN_45S/MKN_45C Ratio | MKN_45S/MKN_45C P Value | MKN_45S/MKN_45C Ratio (KA076TPAc) |
|---|---|---|---|---|
| P60953 (CDC42) | AVK(Ac)YVECSALTQK | 0.82 | 0.45 | 0.58 |
| P30101 (PDIA3) | GIVPLAK(Ac)VDCTANTNTCNK | 0.20 | 0.14 | 0.47 |
| P30101 (PDIA3) | VDCTANTNTCNK(Ac)YGVSGYPTLK | 0.71 | 0.29 | 0.42 |
| P30101 (PDIA3) | EATNPPVIQEEK(Ac)PK | 1.97 | 0.00 | 1.63 |
| P29034 (S100A2) | YSCQEGDK(Ac)FK | 0.15 | 0.09 | 0.44 |
| P00558 (PGK1) | DVLFLK(Ac)DCVGPEVEK | 0.32 | 0.28 | 0.42 |
| P31946 (YWHA) | VISSIEQK(Ac)TER | 1.44 | 0.05 | 1.53 |
| P31040 (SDHA) | AK(Ac)NTVVATGGYGR | 1.09 | 0.29 | 1.83 |
| P34897 (SHMT2) | LNPK(Ac)TGLIDYNQLALTAR | 1.37 | 0.08 | 1.55 |
| Q9NT62 (ATG3) | AK(Ac)TDAGGEDAILQTR | 0.59 | 0.44 | 0.43 |
| Q7L014 (DDX46) | SSGFSGK(Ac)GFK | 17.28 | 0.00 | 2.16 |
| P17844 (DDX5) | K(Ac)FGNPGEK | 6.18 | 0.00 | 1.57 |
| P40939 (HADH) | GFYIYQEGVK(Ac)R | 0.18 | 0.00 | 0.49 |
| O75369 (FLNB) | AFVGQK(Ac)SSFLVDCSK | 0.85 | 0.47 | 0.63 |
| Q08211 (DHX9) | GEEPGK(Ac)SCGYSVR | 0.24 | 0.16 | 0.50 |

treatment, mainly located in the cytoplasm, the nucleus, and mitochondria, functioned primarily in molecular binding, rRNA binding, and chemoattractant activity. Mainly taking part in cell morphogenesis involved in differentiation, regulation of gene expression, *et al.*

Furthermore, through KEGG pathway analysis and PPI network, it was discovered that focal adhesion TGF-beta signaling pathway but anoate metabolism, N-Glycan biosynthesis, mTOR signaling pathway, and adherens junction were the most prominent pathways enriched in ACBP-Se treatment. It was shown that PCBP1, SHMT2, ITGB4 (Integrin beta-4), PGK1, and PDIA3 might be crucial proteins in regulating the process of ACBP-Se treatment. The profile of acetylated proteins might primarily improve our understanding of the role of acetylated proteins in the ACBP-Se treatment of GC.

During the past 30 years, the biological roles of Lys acetylation have been established in non-histone proteins (*Drazic et al., 2016*). Among the differentially expressed acetylated histone proteins between the control group and the ACBP-Se treatment group, SHMT2 K200 was used to show the degree of upregulation, whereas in ACBP-Se treatment.

Serine hydroxymethyl-transferase 2 (SHMT2) is a normal metabolic enzyme that can catalyze the conversion of serine to glycine and produce a living one-carbon unit (*Marrocco et al., 2019*). It supports the synthesis of S-adenosylmethionine (the most crucial direct methyl donor *in vivo*). It provides the necessary precursor for synthesizing proteins and nucleic acids for tumor growth. SHMT2 is highly expressed in various tumors and is associated with a poor prognosis (*Liu et al., 2019*). It has been shown in recent studies that the occurrence and development of tumors, such as acetylation and succinylation, are affected by SHMT2 epigenetic modification (*Yang et al., 2018*). It has been found in recent studies that the SIRT3 deacetylation of SHMT2 can promote the occurrence of colorectal cancer. SIRT3, a major mitochondrial deacetylase, can directly bind to SHMT2, a critical

metabolic enzyme in the one-carbon unit metabolic pathway under glucose starvation. Such deacetylase can also remove the acetylation modification of SHMT2 K95, stabilize the intracellular expression of SHMT2, and maintain the high activity of SHMT2.

We found that SIRT3 and SHMT2 were expressed simultaneously in colorectal cancer patients and that the acetylation of SHMT2 K95 was downregulated. Patients with a high expression of SIRT3 or SHMT2 have a low overall survival rate and a poor prognosis (*Wei et al., 2018*). Acetylation of SHMT2 K95 promoted the binding of E3 ligase trim21 and resulted in the degradation of acetylation of SHMT2 *via* the K63 poly ubiquity in-dependent macroautophagy pathway. This modification also weakened the proliferation rate and tumorigenicity of colorectal cancer cells. It has been suggested in recent studies that SHMT2 may be a feasible drug target for cancer treatment.

Phosphoglycerate kinase 1 (PGK1) is a critical metabolic enzyme in the glycolysis pathway (*Xu, Li & Lu, 2017*). It catalyzes the conversion of 1, 3-bisphosphoglycerate (1, 3-bpg) to 3-phosphoglycerate (3-PG) and produces the initial ATP in the glycolysis pathway. PGK1 is highly expressed in a variety of cancer cells (*Li, Zheng & Lu, 2016*). Topical studies have shown that PGK1 is closely associated with the occurrence and development of tumors. PGK1 is phosphorylated, and acetylation occurs at multiple sites. Translocation of mitochondria and nucleus occurs under specific conditions, directly or indirectly enhancing glycolysis activity and improving the proliferation and growth of cancer cells (*Mazzoni et al., 2016*; *Qian et al., 2019*; *Shuwei & Regen, 1994*). The key acetylation site of PGK1 is k323. The acetylation modification of this site can enhance the activity of the metabolic enzyme PGK1, promote glucose absorption, improve metabolic efficiency, and enhance the tumor-promoting ability of PGK1. The acetylation of the PGK1-k323 site is remarkably correlated with the degree of malignancy.

Still, there is a lack of reports on what and how non-histone acetylation impacts the metastasis of ACBP-Se treatment GC. We showed that of all differentially expressed acetylation. SHMT2 K200 downregulation and PGK1 K97 upregulation may be essential in ACBP-Se treatment. These two acetylated non-histone proteins might be the key regulators of ACBP-Se treatment of GC.

## CONCLUSIONS

In conclusion, compared with the control group, we identified 4,467 acetylation sites from 1,777 proteins in the ACBP-Se treatment. The characterization of acetylation indicated that acetylated proteins might be pivotal in the biological process, molecular binding, and metabolic pathways in the progress of ACBP-Se treatment. Furthermore, by comparing differentially acetylated proteins in the ACBP-Se treatment, we verified that 655 acetylation sites of 499 proteins were downregulated in the ACBP-Se group compared to those in the control group. In comparison, 297 acetylation sites of 238 proteins were unregulated. PGK1 and SHMT2 were the main non-histone acetylations of the most apparent alterations in such sites. These results provide an expanded understanding of acetylation in ACBP-Se treatment and may showed new light on GC molecular-targeting therapy treatment.

## ACKNOWLEDGEMENTS

We'd like to show our great appreciation to the National Natural Science Foundation of China, and Jingjie PTM Biolab (Hangzhou) Co. Inc. for its technical support of proteomics.

### Funding

This work was supported by grant from the National Natural Science Foundation of China (grant no. 81960560) and the Foundation of Inner Mongolia Science and Technology Achievement Transformation (grant no. CGZH2018149), the Natural Science Foundation of Inner Mongolia (grant nos. 2021BS08011 and 2021LHMS08045), Innovation and entrepreneurship training program for college students of Inner Mongolia Medical University (grant no. 202210132025), Laboratory Open Fund Project of Inner Mongolia Medical University (grant no. 2022ZN13), and the Scientific research project of Inner Mongolia Medical University (grant no. YKD2022MS027). The funders had no role in study design, data collection and analysis, decision to publish, or preparation of the manuscript.

### Grant Disclosures

The following grant information was disclosed by the authors:
The Natural Science Foundation of China: 81960560.
Foundation of Inner Mongolia Science and Technology Achievement Transformation: CGZH2018149.
The Natural Science Foundation of Inner Mongolia: 2021BS08011, 2021LHMS08045.
Innovation and entrepreneurship training program for college students of Inner Mongolia Medical University: 202210132025.
Laboratory Open Fund Project of Inner Mongolia Medical University: 2022ZN13.
Scientific research project of Inner Mongolia Medical University: YKD2022MS027.

### Competing Interests

The authors declare there are no competing interests.

### Author Contributions

- Yanan Xu conceived and designed the experiments, performed the experiments, analyzed the data, prepared figures and/or tables, authored or reviewed drafts of the article, and approved the final draft.
- Jianxun Wen conceived and designed the experiments, performed the experiments, prepared figures and/or tables, authored or reviewed drafts of the article, and approved the final draft.
- Wenyan Han analyzed the data, prepared figures and/or tables, and approved the final draft.
- Jin Yan analyzed the data, authored or reviewed drafts of the article, performed the language editing and data statistics, and approved the final draft.

- Wei Jia performed the experiments, prepared figures and/or tables, authored or reviewed drafts of the article, and approved the final draft.
- Xiulan Su conceived and designed the experiments, analyzed the data, authored or reviewed drafts of the article, conceived and designed the experiments, and approved the final draft.

## Data Availability

The raw data is available in the Supplemental Files.

## Supplemental Information

Supplemental information for this article can be found online at http://dx.doi.org/10.7717/peerj.14384#supplemental-information.

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
