# Peer review of "Differential expression of lysine acetylation proteins in gastric cancer treated with a new antitumor agent bioactive peptide chelate selenium"

_PeerJ, doi:10.7717/peerj.14384_

## Round 0.1 · original submission · Major Revisions

Please revise the manuscript and resubmit at the earliest.

Reviewer 1 ·

Basic reporting

1, In order to eliminate language mistakes and inaccuracies, the authors should ask for help by a professional editing service.
2, The structure of ‘results’ section was disorganized, and the author need to merge the figures and reorganize the structure of the article.

Experimental design

1, The gastric cancer cells were treated with 5mg/mL ACBP-se for 24h. The authors should explain the concentration selection.
2, The authors mentioned that ACBP and se were helpful in preclinical gastric cancer treatment, but whether ACBP-se could affect gastric cancer treatment seemed not known? The authors should at least make a simple verification in this article.

Validity of the findings

1, We hope that the authors examine whether ACBP-Se affect the expressions or functions of SHMT2 and PGK1 in gastric cancer cells.

Additional comments

1, Were there any relationships between the changes of acetylation and the phenotypic changes of the gastric cancer cells after the treatment of ACBP-se?

Reviewer 2 ·

Basic reporting

1、The English language should be improved by a fluent speaker.
2、The figures were too fragmented and should be merged.

Experimental design

3、The authors previously reported anticancer activity of peptide‑chelated selenium in vitro, but why authors chose lysine acetylation for the further study? There seemed no evidence that peptide‑chelated selenium treated gastric cancer cells were correlated with k-Ac?

Validity of the findings

4、The authors showed that SHMT2 K200 and PGK1 K97 are the most significant differentially expressed lysine-acetylation sites in ACBP-Se treated cells. But the data from LC-MS/MS could be false positives, could the authors prove it by WB (with custom antibodies?) or at least examin the total acetylation of the two proteins?

Additional comments

5、In this work, the authors report an acetylation-modified-proteomic analysis in gastric MKN-45 cells that were treated with ACBP-Se. Were lysine-acetylation or other PTMs correlated with the progression of gastric cancer? The authors should include these in the discussion.

---

## Round 0.2 · Minor Revisions

Few issues were missed during the revision as indicated by the reviewers. Please revise as per their feedback. Thanks

Reviewer 1 ·

Basic reporting

1、There are several mistakes in the paper, such as the missing of the method of 'TMT labeling' and missing of some spaces between words (line 105, 117, 124, 337, and etc).

Experimental design

The authors have responded to all my questions in these section.

Validity of the findings

1、Though the acetylation of SHMT2 and PGK1 were changed, Whether ACBP-Se affected the expressions or functions of SHMT2 and PGK1 in gastric cancer cells were not known.

Additional comments

The authors added some discussions between the kac of PGK1, SHMT2 and tumor progression.

Reviewer 2 ·

Basic reporting

1、Though the authors said that the languages had been improved by professional editing service, there were still some mistakes in the manuscript. For example, ‘in0.1%’ in line 124 should be ‘in 0.1%’ and ‘tumor igenicity’ in line 337 should be ‘tumorigenicity’.
2、The figures are still fragmented.

Experimental design

1、The authors showed that ACBP-se induced more abundant pan-kac in gastric cancer cells, these should be included in the 'results' section.

Validity of the findings

1、The authors did not confirm the results of LC-MS/MS by another experiments.
2、The discussion of the relationships between Kac and gastric cancer were added in the manuscript.

---

## Round 0.3 · accepted · Accept

The paper can be accepted in its present form based on the revisions.